# Why would individuals use indoor positioning systems? A study from a potential user interface perspective

Thomas Paetow[1,2], Johannes Wichmann[2*], Michael Leyer[2,3]

1 Wismar University of Applied Sciences, Philipp-Müller-Straße 14, Wismar, Germany, 2 Philipps-University Marburg, Barfüßertor 2, Marburg, Germany, 3 School of Management, Queensland University of Technology, Brisbane, Australia

* johannes.wichmann@wiwi.uni-marburg.de

## Abstract

Indoor positioning systems (IPS) differ from outdoor navigation in technological foundations and typical use cases, yet real-world deployments remain relatively rare. For an early-stage acceptance evaluation, literature-derived IPS functions were implemented as an interactive, task-based UI mock-up for a university setting. UTAUT2 was assessed with a pooled sample of 181 potential users across three independent data collections using PLS-SEM with one-tailed BCa bootstrap confidence intervals (90%; 5%/95% bounds). Performance expectancy showed the largest positive association with behavioral intention ($\beta = 0.585$, CI [0.480; 0.678]). Social influence ($\beta = 0.179$, CI [0.115; 0.245]), price value under a communicated one-time fee of €0.99 ($\beta = 0.179$, CI [0.096; 0.260]), and habit ($\beta = 0.179$, CI [0.097; 0.265]) also related positively to intention ($R^2 = 0.781$). Effort expectancy ($\beta = -0.051$, CI [−0.136; 0.023]), facilitating conditions ($\beta = -0.027$, CI [−0.108; 0.046]), and hedonic motivation ($\beta = -0.002$, CI [−0.109; 0.113]) were not supported; a robustness model indicated a negative age × price value interaction ($\beta = -0.164$, CI [−0.314; −0.071]). The findings extend UTAUT2 evidence to prototype-based IPS UI evaluation and suggest that performance-related beliefs dominate early intention formation. Practically, prototype concepts should prioritize and clearly communicate features that reduce uncertainty, time cost, and search effort in complex indoor environments. Because the stimulus was a UI mock-up and the outcome was self-reported intention, IPS-specific technical frictions (e.g., localization inaccuracies, latency, route deviations) and sustained real-world use were not observed, which may limit transferability to deployed IPS.

## 1 Introduction

Location-based services (LBS) and GPS-based systems are popular for outdoor navigation [1]. While outdoor navigation is relatively mature, indoor navigation is still under development because GPS does not work indoors, and alternative

**Data availability statement:** Our data and Appendices A to G are available here: https://osf.io/f9t4h

**Funding:** The author(s) received no specific funding for this work.

**Competing interests:** The authors have declared that no competing interests exist.

technologies must be evaluated to enable reliable indoor positioning and navigation [2,3]. Indoor navigation can, however, lead to significant savings in time and resources [2], especially when individuals navigate in complex building structures like universities (e.g., [4,5]), airports (e.g., [6]), shopping malls (e.g., [7]), or hospitals (e.g., [8]). While indoor navigation has received increasing attention in recent years [3], indoor and outdoor navigation differ in key aspects. Indoor navigation relies on different technologies and wayfinding approaches [2,5]. Relevant use cases also differ across settings (e.g., [9]). Accordingly, users may have different reasons for using indoor navigation than for outdoor navigation, because indoor navigation is typically used in Global Navigation Satellite System (GNSS)-denied, multi-floor environments and often involves context-specific tasks such as locating rooms or people and finding indoor services (e.g., restrooms, copiers, cafeteria stations) [10–12]. Even if IPS capabilities are available, the potential value of the service depends on users' intention to adopt it and to continue using it over time [13,14]. Therefore, understanding the determinants of behavioral intention is a key research problem beyond purely technological feasibility. Consistent with this view, IPS research has predominantly emphasized technological aspects such as positioning technologies and accuracy, whereas evidence on adoption drivers and behavioral intention remains comparatively limited, especially when evaluations are grounded in interaction with user interface (UI) mock-ups rather than in the general idea of IPS alone [2,15,16]. Studies that examine individuals' reasons for typical IPS functions mostly address IPS either at an abstract or technological level and do not investigate UI mock-ups [2,15,16]. For potential users, the structure of an interface is often more important than the technical details in the background [17].

Against this background, this study addresses two research questions: RQ1: How should an IPS UI mock-up be structured to provide relevant information for indoor navigation according to related work? RQ2: Which UTAUT2 factors are associated with potential users' behavioral intention to use such an IPS UI mock-up after interaction?

As the study is based on an interactive UI mock-up rather than a functional IPS, IPS-specific technical frictions such as localization inaccuracies, latency, or navigation errors are not present, which may upward-bias intention compared to real deployments. To address the research questions, IPS functionality is grounded in literature-based use cases and operationalized in an actionable UI mock-up, enabling potential users to form adoption-relevant evaluations based on concrete interaction rather than an abstract scenario alone. Such a mock-up anchors respondents' evaluations in concrete task experience and may yield more informed intention judgments than a purely abstract scenario-based survey [18]. At the same time, intentions can diverge from subsequent behavior, and the present study cannot be directly related to sustained real-world usage, post-adoption dynamics, or behavioral consequences. Accordingly, the study should be interpreted as an early-stage, prototype-based evaluation of determinants of behavioral intention toward an IPS UI mock-up rather than evidence on the adoption of fully functional IPS deployments. Therefore, an IPS

mock-up is developed and made actionable via task-based interaction paths that reflect typical indoor navigation functions based on conceptual ideas of indoor navigation design (e.g., [7,12]).

As an application context for specific tasks within the IPS mock-up, universities are used because their building structure is often complex and they are frequented by students but also visitors, e.g., for scientific conferences [6]. Further, universities contain use cases such as waiting times in the cafeteria or finding a free copy machine [10]. As such, insights from investigating these use cases can be applied to other IPS application scenarios, as they are conceptually similar. For instance, IPS can be used to determine unused hospital beds or to check waiting times at security checkpoints in airports or hospitals [2,15,16].

To explain behavioral intention toward IPS from a potential user perspective, the evaluation is grounded in the Unified Theory of Acceptance and Use of Technology 2 (UTAUT2) by Venkatesh et al. [13]. UTAUT2 is particularly suitable because IPS usage is voluntary and service-like, and UTAUT2 explicitly incorporates adoption drivers relevant to such contexts (e.g., habit) in addition to performance expectancy (see also related UTAUT2 applications in LBS contexts, e.g., [19]). The study contributes theoretically by (i) contextualizing UTAUT2 to the IPS UI domain with actionable, use-case-driven functionality, (ii) linking a task-based mock-up interaction setting to adoption-relevant perceptions to reduce reliance on purely abstract judgments, and (iii) providing empirical evidence on which UTAUT2 determinants are most salient for IPS UI adoption in complex indoor environments. In addition, the study contributes practically by deriving implications for IPS feature prioritization and service positioning in contexts such as universities.

The paper is organized as follows: Section 2 starts with the theoretical background of IPS. Section 3 then develops the IPS UI mock-up based on the theoretical background that follows general guidelines and represents relevant typical functions for indoor navigation. Section 4 presents UTAUT2 to derive the research model with hypotheses for evaluation purposes. Section 5 describes the method, including tasks within the mock-up and the study procedure. Section 6 presents the results that are discussed in Section 7. Finally, Section 8 concludes with theoretical and practical implications and describes the limitations of this research and future research.

## 2 Research on behavioral aspects of navigation systems

### 2.1 Behavioral aspects of indoor positioning systems

Indoor positioning systems (IPS) utilize algorithms to determine the real-time location of individuals or objects within buildings or complexes [20]. For IPS, various technologies such as Wi-Fi, Bluetooth Low Energy (BLE), Radio Frequency Identification (RFID), or ultrasound are employed, using techniques such as triangulation or trilateration [8,21]. For further information concerning IPS from a technical perspective, the comprehensive survey of Zafari et al. [2] or García-Catalá et al. [22] is recommended.

To develop an IPS that meets stakeholders' demands, use cases and prototypes must be designed [2]. For instance, Bucheli Fuentes et al. [23] employed the evolutionary development model [24] to construct a BLE-based IPS prototype that used the self-improving technique called fingerprinting (see [2]) and evaluated its accuracy. Alattas et al. [4] created a combined usage model of IndoorGML and the Land Administration Domain Model. University buildings were used as examples to show relationships between user types and spaces that include rights, restrictions, and responsibilities, such as access times to lecture halls, in order to support indoor navigation. Karabtcev et al. [5] investigated the calculation of an efficient number of BLE beacons and their effective placement in a building, using the example of the Campus of Kemerovo State University. Delnevo et al. [9] also presented a BLE-based system for indoor wayfinding in their study called "AlmaWhere", which is specifically aimed at the use case of disabled individuals. In their study, Yee Tan et al. [7] introduce an IPS for a shopping mall that operates using Wi-Fi and, based on developed use cases, allows users to navigate while shopping. In addition, it provides additional functions such as searching through a UI, providing comprehensive information about shops, and additional information such as shop promotions [7]. Further, Zafari et al. [2] states that considering UIs for sufficient IPS development is important. For route-planning in complex indoor environments, van Schaik et al. [25]

conducted experiments and showed that 3D elements are important and should be implemented in indoor wayfinding, as the acceptance of such route-planning was high. This proposition is also supported by Mukawa et al. [26], who performed experiments to investigate how augmented reality (AR) approaches influence indoor scene memorability. They determined that augmented reality approaches are indeed helpful but should be enriched with navigation information, as performed by van Schaik et al. [25].

While several studies have evaluated IPS UI design for hospitals (e.g., [27]), shopping malls (e.g., [7]), and universities (e.g., [10,11]), most of these studies only consider technology aspects, as highlighted by Liu et al. [16] and Zafari et al. [2], without examining factors such as intention to use. Wichmann and Leyer [15] investigated the intention to use IPS in hospitals using the Reasoned Action Approach (RAA), according to Fishbein and Ajzen [18]. However, they did not evaluate UIs or mock-ups. Therefore, research is needed combining use cases and prototype evaluation on one hand and intention to use on the other [17]. Accordingly, combining use-case-driven prototyping with adoption research remains an important gap.

### 2.2 Behavioral aspects of outdoor positioning systems

Whereas related work on outdoor positioning systems (OPS) investigates approaches using the Technology Acceptance Model (TAM) or UTAUT, evidence remains limited on user interface mock-ups of such systems. Moreover, indoor navigation differs from outdoor navigation in constraints and use cases, so findings from OPS cannot be transferred directly to IPS UI concepts [3,5,9]. To investigate factors impacting the use of navigation systems by Chinese private and professional drivers, Ge et al. [28] used TAM, which they extended using trust. Their results indicate that perceived usefulness, perceived ease of use, and attitude toward the system acted as important mediators. In their study investigating drivers' acceptance of car navigation systems, Park and Kim [29] extended TAM using variables such as perceived locational accuracy, perceived processing speed, service and display quality, and satisfaction. Their model shows that perceived processing speed and locational accuracy of car navigation systems were their key psychological constructs, while satisfaction played a moderate role. In their study investigating the adoption of augmented reality wayfinding in a heritage tourism context, Wang et al. [30] used UTAUT. Their results show that the effort expectancy outweighs performance expectancy, and that trust and self-efficacy are important predictors for the adoption of the wayfinding system.

## 3 Developing an IPS UI mock-up

### 3.1 Use cases and related functions for IPS

To develop an IPS UI mock-up, use cases are first generated based on prior literature to capture real-world scenarios (see Table 1 for an overview). The six use cases cover the main functions of IPS and are explained in the following.

**Table 1. Use cases for the IPS UI mock-up.**

| No. | Use cases | References |
|-----|-----------|------------|
| 1 | Searching and navigating within complex buildings | [7,11,12] |
| 2 | Exploration of buildings using a digital building map | [7,10,31] |
| 3 | Searching and locating objects within buildings | [11,31–33] |
| 4 | Locating free parking spaces near the destination of indoor navigation | [10,12] |
| 5 | Real-time occupancy at specific locations that affect indoor navigation | [10,34,35] |
| 6 | Appointment scheduling | [10,34] |

The first use case addresses navigating in complex buildings, which requires searching and finding specific locations, similar to outdoor navigation for pedestrians, such as automated teller machines (ATMs) [36]. Indoor navigation should therefore be able to navigate to specific rooms, people, and points of interest (POIs) (e.g., [11]), such as the nearest restroom in a university building [12] or the nearest electronics store in a shopping mall [7]. Accordingly, a function for searching and navigating within complex buildings is derived.

The second use case addresses exploring and orienting the environment using a map, as is the case with outdoor navigation [36]. Additionally, various POIs within buildings, such as shops [7], artworks [31] or copiers [10], are referenced that can be displayed. Indoor navigation also requires floor-specific building views. Some outdoor navigation services provide venue-specific indoor maps (e.g., for airports), but GNSS-based outdoor navigation does not provide reliable indoor positioning and routing across multiple floors. Consequently, indoor navigation requires dedicated indoor positioning and floor-aware map representations [3], which is also one of the major differences between indoor and outdoor navigation. Accordingly, a function is derived that enables the exploration of buildings using a digital building map.

The third use case represents locating specific objects within buildings, such as hospital beds [32] or newborns tracked using wearables in hospitals [33]. In a university context, this can also include locating books in a library [11] and, in museums, locating artworks [31]. Accordingly, a function is derived for searching and locating objects within buildings.

The fourth use case represents the scenario where an individual needs a parking space near the destination [10,12]. In the context of indoor navigation, free parking spaces for electric vehicles located at the respective buildings must be found. Therefore, a function is derived for locating free parking spaces near the desired destination.

The fifth use case addresses the scenario in which certain events can delay indoor navigation, such as waiting in line at a cafeteria [10] or in an emergency room [34], as well as waiting at security checkpoints at airports [35]. Thus, a function is derived that reflects real-time occupancy at specific locations that affect indoor navigation and arrival time.

The sixth use case assumes that navigation can be initiated based on appointments, such as doctors' appointments in hospitals [34], lecture times, or appointments with colleagues at a university [10]. Furthermore, it is derived that booking specific rooms as appointments should also be enabled, such as booking rooms for learning or an operating room. Thus, a function is derived that implements appointment scheduling in the IPS.

### 3.2 Functions within the IPS UI mock-up

After deriving the use cases for IPS in general, an IPS UI mock-up for universities containing IPS functions was designed. An iterative approach based on rapid (mock-up) prototyping [37] was followed in developing the IPS UI mock-up. The IPS UI mock-up, developed as a mobile application for smartphones, offers six functions, as shown in Fig 1 (Main Screen): (i) search, (ii) map, (iii) library, (iv) parking search, (v) planner, and (vi) cafeteria. Based on research regarding IPS design for universities (e.g., [10,11]), the inclusion of a search function is essential for locating specific rooms or points of interest, like the nearest restroom [12], before starting a navigation task.

For the navigation process, the augmented reality approach was selected, which is applicable in the context of indoor navigation for, e.g., hospitals [27], universities [38], airports [35], office buildings [26], and shopping malls [39]. Recent evidence from AR wayfinding research indicates that ease of use (effort expectancy) can be more salient than performance-oriented benefits and that perceived interactivity supports intention to use, supporting a design that prioritizes clear, low-friction AR guidance and interaction [30]. By using this augmented reality approach and implementing real-world navigation information, prior work is also considered that points to the effectiveness of navigational memory and processes when such information is provided [25,26]. In addition, a 2D map is used for displaying the search results, which is deemed mandatory for IPS [10,11], as it is for pedestrian outdoor navigation systems [36].

Through rapid prototyping, it was determined that implementing different views for various floors is necessary, something GNSS-based outdoor navigation cannot provide reliably indoors (especially across floors) [2]. Regarding the university-specific application scenario, a mock-up function was integrated that allows users to search for specific items,

**Fig 1. IPS UI mock-up (f.l.t.r.)**: Main Screen, Search Function, Map Function, Library Function.

as proposed by previous research for IPS for universities [12]. According to Hadwan et al. [11], the example of a library was used to find books. According to Hammadi et al. [12], the mock-up also has an integrated parking function, which shows free parking lots nearby in real time. Further, a planner function was integrated for appointments with university members and to book free rooms, according to Paetow et al. [10]. Additionally, the developed mock-up provides additional information, such as office hours of university members, as well as their contact details, as recommended by Hammadi [12]. Ultimately, a cafeteria function was implemented in the IPS that helps users to plan their lunch breaks effectively, as this function provides real-time occupancy information for each food station, as recommended by Paetow et al. [10] and Hammadi et al. [12]. Further, visits to the cafeteria are eased by providing information about daily menus via the IPS.

It was also determined that other prototypes or mock-ups also included functions such as "settings" or "favorites" [10,12]. Specifically, for frequently visited locations such as POIs like the nearest restroom, a shortcut for "favorites" was included to provide quick access. The "settings" function is also considered necessary and was implemented, not only for the possibility of turning on, for example, push notifications, but especially regarding accessibility, according to Delnevo et al. [9]. In this regard, a function was integrated that enables specific routes to be taken for people with disabilities and the ability to turn the speech output on and off. In the mock-up, these functions are subsumed as general functions. Based on these findings, functions were then derived that were integrated into the IPS mock-up as a graphical user interface using Adobe XD.

## 4 Theory for evaluation

### 4.1 The unified theory of acceptance and use of technology

There are several models that investigate the user's behavioral intention to use new technologies. Some of the models (e.g., [40,41]) are based on the TAM [42] and others (e.g., [43,44]) are based on the Theory of Planned Behavior (TPB) [45]. In 2003, Venkatesh et al. [46] introduced UTAUT to investigate the intention and acceptance of new

technologies among various stakeholders, including managers [46]. UTAUT integrates several existing acceptance theories and models, such as TAM [42], Social Cognitive Theory [27], and Innovation Diffusion Theory [47], to determine the factors that relate to the intention to use new technology. UTAUT also provides insights into the underlying reasons for this intention. Four constructs were identified as determinants of behavioral intention and usage behavior in UTAUT: performance expectancy (PE), effort expectancy (EE), social influence (SI), and facilitating conditions (FC) [46]. In the meanwhile, UTAUT has become a widely used theoretical framework for information system adoption [48], used in various domains, such as implementing electronic health records in hospitals [49], food delivery services applications [50], and mobile payment applications [51].

In 2012, Venkatesh et al. revised their original UTAUT model [13]. They introduced UTAUT2, an extension of the UTAUT model created to evaluate new technology in the consumer market [13]. UTAUT2 builds on the strengths of the UTAUT model. It integrates new predictors that allow a more comprehensive examination of behavioral intention to use new technologies, such as hedonic motivation (HM), price value (PV), and habit (HT) [13]. Since UTAUT2 has been widely used in the consumer market and for LBS services (e.g., [19,52]), UTAUT2 is used for this study. As the IPS is a UI prototype according to Tamilmani et al. [48], this study is in the early adoption phase. Hence, as UTAUT2 is particularly suitable for investigations of IS adoptions in early phases according to Tamilmani et al. [48], UTAUT2 is used to do so. Further, several meta-studies confirm that UTAUT2 is sufficient for technology adoption (e.g., [48]). It is also applied in this research to investigate the behavioral intention to use IPS and address the research question.

## 4.2  Behavioral intention to use and location-based services

In the last years, there has been increasing research interest in indoor positioning systems (IPS) for various applications, including airports, libraries, shopping malls, hospitals, and universities (e.g., [2,6–8,10,53]). Applications in practice are, however, rare [53]. Furthermore, LBS are increasingly important since the number of mobile applications evaluated using UTAUT is rising [48]. As such, LBS use cases are important for the development of IPS use cases, as they are conceptually similar, primarily aiming to determine the spatial position of users or objects in indoor or outdoor environments and provide location-based information, helping users navigate unfamiliar surroundings (e.g., [2,3]).

Ayuning Budi et al. [52] applied the UTAUT model to explain the use of location-based applications in emergencies, specifically the intention to use a "panic button" in their study. They found that UTAUT is applicable in this context. They identified performance expectancy as a key driver for using LBS, such as a panic button in emergencies.

Performance expectancy, facilitating conditions, and effort expectancy have been identified as the three main drivers for the intention to use of LBS in tourism contexts (19,54). Additionally, Uphaus et al. [54] demonstrated that hedonic motivation is important for the intention to use LBS in tourism contexts. Gupta and Dogra [19] also showed that users' habits are important to consider in the intention to use LBS, specifically "mapping apps" during vacations. Hence, performance expectancy, facilitating conditions, effort expectancy, and hedonic motivation are important factors to consider in the intentional use of LBS in tourism contexts [19,54].

These studies collectively support the applicability of UTAUT and UTAUT2 models in the context of LBS. Therefore, these studies show that UTAUT and UTAUT2 are applicable in the context of location-based services. Furthermore, UTAUT2 has been analyzed in the context of LBS with meta-studies (e.g., [17]). However, in the field of location-based services, the use cases of outdoor navigation are different from indoor navigation (e.g., [7,9]), so transferring the results of those two areas is not sufficient.

In addition to UTAUT and UTAUT2, there is further research available on other theories that investigate the behavioral intention to use in this domain. For example, Wichmann and Leyer [55] used the RAA and found that social norms significantly impact the behavioral intention to use IPS in a hospital, meaning if family and friends recommend such a system.

 

However, research on UTAUT2 and location-based applications like IPS, including use cases and an IPS UI mock-ups, could not be identified. Thus, this gap is addressed by evaluating an IPS UI mock-up containing functions based on use cases.

## 4.3 Hypotheses and research model

Effort expectancy, which refers to the perceived ease of using a technology, has been shown to positively relate to behavioral intention toward LBS usage, according to Gupta and Dogra [19] and Uphaus et al. [54]. Chen and Tsai [47] also identified perceived ease of use, equivalent to effort expectancy, as a positive predictor of LBS usage intention. Thus, the first hypothesis is:

H1: Effort expectancy positively relates to an individual's intention to use an IPS UI mock-up.

Venkatesh et al. [13] define performance expectancy as the extent to which the technology in question helps consumers in performing specific activities. It is an important predictor in LBS adoption studies. Gupta and Dogra [19] found that performance expectancy positively impacts the behavioral intention to use LBS, supported by other studies (e.g., [56,57]) indicating it as a key factor influencing LBS usage. Further, Chen & Tsai [47] determined that perceived usefulness positively relates to the intention to use LBS. Hence, the second hypothesis is:

H2: Performance expectancy positively relates to an individual's intention to use an IPS UI mock-up.

Venkatesh et al. [13] define social influence as the extent to which an individual perceives that peers would appreciate the individual using the new system. In this study, peers are best friends, family, colleagues, and superiors, who are common peers in UTAUT studies [18,19,58]. Related work determined that social influence positively relates to the intention to use LBS [19] and IPS in hospitals [15]. Thus, the third hypothesis is:

H3: Social influence positively relates to an individual's intention to use an IPS UI mock-up.

Facilitating conditions are consumers' perceptions of their own resources and their ability to support performing a behavior [58]. UTAUT2 proposes that an individual's perception of facilitating conditions directly affects technology acceptance, which can be supported or hindered by their surrounding environment, e.g., availability of resources, technical infrastructure, and support [13]. Hence, the fourth hypothesis is:

H4: Facilitating conditions positively relate to an individual's intention to use an IPS UI mock-up.

Venkatesh et al. [13] define hedonic motivation as the pleasure or enjoyment individuals feel by using the new system. Brown and Venkatesh [58] stated that hedonic motivation is a crucial predictor of technology adoption and usage. Concerning LBS for tourism contexts, Gupta and Dogra [19] and Uphaus et al. [54] ascertained that hedonic motivation positively affects an individual's behavioral intention. Thus, the fifth hypothesis is:

H5: Hedonic motivation positively relates to an individual's intention to use an IPS UI mock-up.

In UTAUT2, Venkatesh et al. [13] state that the perceived price value is crucial for determining behavioral intentions, particularly if consumers have to pay for the new system. When consumers perceive that the benefits of the technology outweigh the costs, the price value becomes positive and positively affects behavioral intentions [13]. Because price value is conceptualized as a benefit–cost trade-off, it depends on the communicated pricing scheme [13]. Accordingly, price value is hypothesized under the specific pricing frame used in the evaluation. Gupta and Dogra [19] also find that price value positively relates to behavioral intentions for LBS. Hence, the sixth hypothesis is:

H6: Price value positively relates to an individual's intention to use an IPS UI mock-up.

Habit is the result of past behavior and experiences, according to Venkatesh et al. [13], and is an important antecedent for predicting current actions [59]. For technology acceptance, several studies have shown that past and recurring behavior is necessary (e.g., [60–62]). For location-based services, Gupta and Dogra [19] determined that habit positively relates to behavioral intentions and the actual use of those apps. Thus, the seventh hypothesis is:

H7: Habit positively relates to an individual's intention to use an IPS UI mock-up.

Fig 2 summarizes the resulting research model with the seven hypotheses.

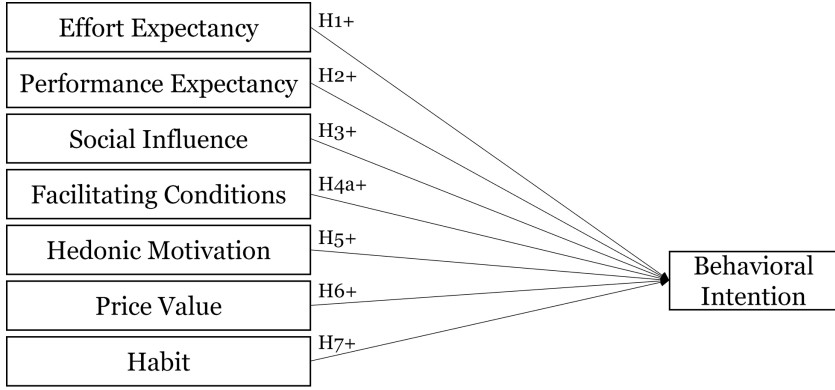

**Fig 2. Research model.**

## 5 Method

### 5.1 Measurements and procedure

Prior to responding to the questionnaire addressing the hypotheses, participants were confronted with a series of use cases comprised of tasks operationalizing the functions of the developed IPS UI mock-up (see Section 5.2). Participants were required to navigate through the IPS UI mock-up (see Appendix E) to accomplish the relevant tasks associated with each use case. Completing each task revealed a code that had to be entered into the questionnaire to proceed to the next task, thereby also serving as attention checks [63]. Participants had to enter all task codes to complete the survey. Otherwise, the survey was terminated.

Afterwards, participants answered a questionnaire (see Appendix A) that was mainly adapted from Venkatesh et al. [13] to the context. Participants responded to each item on a 7-point Likert scale, ranging from "strongly disagree" (1) to "strongly agree" (7). For the price value measurement, the pricing scheme was communicated as a one-time fee of €0.99 (see Appendix), and the construct was measured under this specific pricing frame.

Additionally, the questionnaire included control questions to verify the validity of self-reported university membership to improve statistical power and reduce Type-II errors [63]. If participants were not coherent with their answers regarding university structure and faculty at the beginning and end of the questionnaire, the questionnaire ended for these participants.

### 5.2 Tasks within the IPS UI mock-up

To be able to test the functions of the developed IPS UI mock-up, tasks were designed that participants had to successfully perform in a university context. The IPS UI mock-up was implemented as an interactive prototype enabling task-based navigation across the six core functions.

The (i) search function (see Fig 1) was used to locate Professor John Doe and then initiate navigation to his office. The (ii) map function (see Fig 1) was intended to display the nearest copier based on the current location and provide additional information. The (iii) library function (see Fig 1) was employed to search for a book and display its location. The (iv) parking finder function (see Fig 3) was meant to locate available charging stations for electric cars. The (v) planner function (see Fig 3) was used to reserve an available room for a group of four people, and the (vi) cafeteria function (see Fig 3) was used to display the wait time until the daily special meal is served.

As the IPS is a user interface (UI) mock-up, according to Tamilmani et al. [48], this study is in the early adoption phase. Hence, as UTAUT2 is particularly suitable for investigations of IS adoptions in early phases, according to Tamilmani et al. [48], UTAUT2 is applied to do so.

                                                                

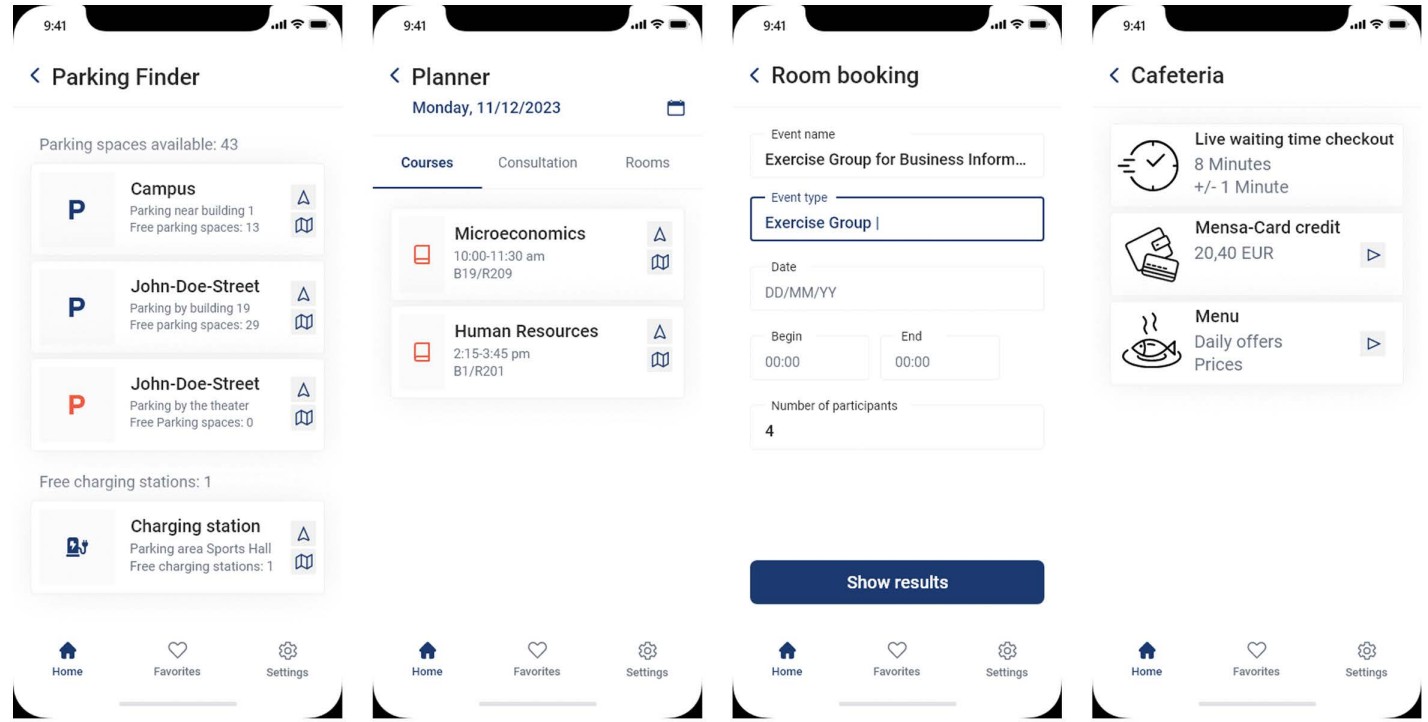

**Fig 3. IPS UI mock-up (f.l.t.r.)**: Parking Finder Function, Planner Function and Room Booking, Cafeteria Function.

### 5.3 Participants and data collection

Data were collected in Germany using an identical questionnaire and study protocol across three independent data collections conducted across two periods (October–November 2022; January 2026). Two samples were recruited via the crowdworking platform Clickworker and were compensated via the platform, whereas one sample was recruited at a university on a voluntary basis. Because monetary incentives can encourage rapid responding, strict, predefined data-quality procedures were implemented, including seven attention checks, which were applied consistently across all three data collections [63].

After a short use-case presentation, participants interacted with the IPS UI mock-up illustrating core IPS functions. To ensure active engagement with the UI mock-up, participants were required to navigate within the mock-up to retrieve a code (see Appendix A, IPS UI Mockup) that enabled continuation to the questionnaire. After completing the prototype task, participants answered the survey items (see Appendix A, Questionnaire).

In the first data collection (Clickworker; October–November 2022), 464 questionnaires were started with 73 participants aborting and 268 responses being excluded because at least one of the seven attention checks was answered incorrectly. This resulted in N = 123 valid responses. The high exclusion rate is common for platforms like Clickworker when strict screening criteria are applied; such criteria aim to reduce careless responding and increase data quality [63]. This first sample comprised 44 female participants (35.8%) and 79 male participants (64.2%), with ages ranging from 18 to 55 years (M = 25.20, SD = 5.05). Most participants were 20–29 years old (85.4%).

In the second data collection (university; January 2026), 67 participants started the survey, 20 participants aborted, and 7 responses were excluded after applying the same data-quality criteria, resulting in N = 40 valid responses. This second sample comprised 19 female participants (47.5%) and 21 male participants (52.5%), with ages ranging from 18 to 33 years (M = 22.73, SD = 3.15), and most participants were 20–29 years old (87.5%).

In the third data collection (Clickworker; January 2026), 21 participants started the survey, and 2 responses were excluded after applying the same data-quality criteria, resulting in N = 18 valid responses. This third sample comprised 8 female participants (44.4%) and 10 male participants (55.6%), with ages ranging from 20 to 30 years (M = 25.56, SD = 2.91), and most participants were 20–29 years old (88.9%).

All samples were pooled for the main analysis because they were collected with the same target group, using identical procedures, and the same measurement instrument. A statistical comparison between the samples using MANOVA (see Appendix G) did not indicate an overall multivariate difference in construct scores; univariate follow-ups suggested a difference only for habit, which was higher in the January 2026 Clickworker sample. The implications are discussed in the Discussion and Limitations sections. The combined dataset comprised N = 181 valid responses, including 71 females (39.2%) and 110 males (60.8%). In the pooled sample, participants were aged 18–55 years (M = 24.69, SD = 4.62), and most were 20–29 years old (86.2%).

## 5.4  Data analysis

The data were analyzed using partial least squares path modeling (PLS-SEM) to explore the hypotheses associated with the research model (see Fig 2). PLS-SEM was chosen because it allows the simultaneous estimation of latent constructs measured by reflective indicators and a formative construct (facilitating conditions), and because it is well-suited for prediction-oriented model assessment [64,65]. Additionally, PLS path modeling is particularly suited for investigating new phenomena, as noted by Chin and Newsted, which aligns with the nature of the research on indoor navigation and its associated applications [66]. To determine effect sizes, $f^2$ effect sizes were calculated following Cohen's recommendation [67]. However, considering the sample sizes, the threshold for small effect sizes was adjusted based on power analyses using G*Power 3.1.9.7 [68], following the approach of Kock and Hadaya [69]. Thus, given seven predictors, the threshold for a small effect was set to $f^2 = 0.05$ (see Appendix C).

The analysis was conducted using SmartPLS (version 4.1.1.6) [70]. The PLS algorithm was estimated using the path weighting scheme. Because several indicators showed pronounced non-normality (e.g., skewness/kurtosis in FC indicators), statistical inference is reported using bias-corrected and accelerated (BCa) bootstrap inference [64]. Given the directional hypotheses derived from UTAUT2 [13], one-tailed tests were applied for hypothesis testing. BCa percentile confidence intervals are reported using the 5%/95% bounds, consistent with the one-tailed testing approach.

According to Hair et al. [64], the evaluation of the measurement model prior to the analysis of the structural model involves a 4-step process. The first step in this evaluation process involves an examination of the loadings of the indicators that are mandatory for establishing the validity and reliability of the reflective measurement model. The indicator reliability values of the reflective variables "performance expectancy", "effort expectancy", "social influence", "hedonic motivation", "price value", "habit", and the dependent construct "behavioral intention" exceeded the recommended threshold of 0.708, indicating adequate indicator reliability [64]. HT2 was slightly below this threshold with a loading of 0.688 but was retained due to its conceptual relevance and satisfactory construct-level quality (see Appendix B, Table 1).

Furthermore, for the formative variable "facilitating conditions (FC)", multicollinearity was assessed by examining the variance inflation factors (VIF). For the formative FC indicators, VIF values were below 5, indicating that multicollinearity was not a critical issue (see Appendix B, Table 4). In line with Hair et al. [64], the relevance and significance of the formative indicators were then assessed using the outer weights and their bootstrap inference. The results show that FC4 (weight = 0.801, p < .001) has a statistically significant outer weight, whereas FC1 (weight = 0.205, p = .265) and FC2 (weight = 0.116, p = .366) have non-significant outer weights (see Appendix B, Table 2). However, FC1 and FC2 were retained because their outer loadings are substantial and statistically significant (FC1 = 0.702, FC2 = 0.708; both p < .001), supporting their absolute contribution and content validity of the formative construct (see Appendix B, Table 1).

The second step is the evaluation of internal consistency reliability [64]. Cronbach's alpha, as well as composite reliability (rho_A and rho_C), were used. Overall, the reflective constructs show very high internal consistency (see Appendix B,

Table 3). In a third step, convergent validity was examined for each reflective construct measure and assessed using the average variance extracted (AVE) [64]. As shown (see Appendix B, Table 3), all AVE values are 0.50 or higher, indicating that more than 50% of the variance of each construct is explained by its indicators [64].

The fourth step [64] of the evaluation process involves assessing discriminant validity to ensure that a construct differs from other constructs in the model. Because the Fornell-Larcker criterion [71] has been shown to be less effective, mainly when a construct's indicator loadings exhibit minor variation, the heterotrait-monotrait ratio (HTMT) of correlations proposed by Henseler et al. [72] was used. All HTMT values were below the threshold of 0.90 (see Appendix B, Table 5), as required by Henseler et al. [72]. In addition, bootstrapped HTMT confidence intervals (BCa) were inspected, supporting discriminant validity as the BCa confidence intervals did not include 1.0 (see Appendix B, Table 5).

As the evaluation of the measurement model is satisfactory, the analysis of the structural model is performed [64]. Prior to structural model assessment, common method bias was examined using the full collinearity assessment proposed by Kock [73]. This procedure involves regressing each latent variable on all other latent variables in the model and inspecting the resulting variance inflation factors (VIFs). According to Kock [73], VIF values exceeding 3.3 indicate pathological collinearity and potential common method bias. In the present study, several full collinearity VIFs exceeded 3.3. However, all values remained below 5.0 (range: 2.48–4.56; see Appendix B, Table 18). Following Kock [73], who notes that higher VIFs may occur in complex models with conceptually related constructs, the results do not indicate severe common method bias.

Evaluating collinearity before analyzing the structural relationships is also important to avoid the potential biasing of regression results. One way to achieve this is by examining the variance inflation factor (VIF) values, ideally below 3 [64]. No value exceeds this threshold (see Appendix B, Table 7). Path coefficients show the estimates of relationships in the structural model, i.e., the hypothesized relationships between constructs [64]. Analyzing the coefficients and their statistical significance allowed the hypotheses to be tested. To determine the significance of the effects, the BCa bootstrapping procedure described above was used to assess whether the path coefficients differ from zero in the hypothesized direction (see Appendix B, Table 7). Furthermore, the $R^2$ value was examined. A guideline for interpreting $R^2$ values suggests that values of 0.75, 0.50, and 0.25 indicate substantial, moderate, and weak explanatory power, respectively [64,74]. The $R^2$ value suggests that the model explains 78.1% ($R^2 = 0.781$, $R^2$ adjusted $= 0.772$, see Appendix B, Table 8-9) of the total variance and can be considered substantial [74,75]. Effect sizes were additionally assessed using $f^2$ to quantify the relative impact of each predictor on BI (see Appendix B, Table 10).

To assess predictive performance, PLSpredict ($Q^2$_predict) was employed. Compared to the naïve linear regression model (LM) benchmark, the model exhibits better predictive performance as it generates lower prediction errors (RMSE/MAE) for the indicators of the target construct (see Appendix B, Tables 12-13). Cross-validated predictive ability test (CVPAT) was also used as an additional benchmark test. CVPAT indicates that PLS-SEM performs significantly better than the indicator-average (IA) benchmark (see Appendix B, Tables 14-15). However, compared to the LM benchmark, the loss differences are smaller and do not reach statistical significance (see Appendix B, Tables 16-17). Finally, SRMR is reported as an approximate model fit index [76]. The SRMR value is 0.088, which is slightly above the conservative 0.08 cut-off proposed by Hu [77]. Therefore, SRMR is interpreted cautiously and primarily rely on the measurement, structural, and predictive assessments (see Appendix B, Table 11).

As a robustness check consistent with UTAUT2, a moderation model including age and gender (direct effects and interaction terms with all predictors) was estimated using the same PLS-SEM settings and BCa bootstrap inference (one-tailed; 5%/95% bounds). In addition, model performance was compared against the main model using $R^2/R^2$adj, SRMR, and PLSpredict/$Q^2$predict. The full moderation output and the performance comparison are reported in Appendix B (Tables 19-20).

## 5.5 Ethical considerations

All procedures performed in this survey involving human participants were in accordance with the ethical standards of the 1964 Helsinki declaration and its later amendments or comparable ethical standards. As stated, the data was collected

with a crowdworker platform (Clickworker) on which participants contribute anonymously and receive a monetary award for their time spent, which fulfilled the minimum wage criteria in Germany. Further, informed consent was obtained from participants that the survey data may be used for research purposes. It was ensured that the data analysis and presentation were exclusively anonymized, and therefore no conclusions can be drawn about certain individuals. Since the data gathering did not involve any personal information and was a non-interventional study, no ethical approval was required according to German national laws.

## 6  Results

The means, standard deviations, and Pearson correlation coefficients between the constructs can be found in Table 2.

The structural model explains 78.1% of the variance in behavioral intention ($R^2 = 0.781$; adjusted $R^2 = 0.772$; Appendix B, Tables 8–9). Hypotheses were evaluated using standardized path coefficients and bias-corrected and accelerated (BCa) bootstrap confidence intervals. Consistent with one-tailed hypothesis tests ($\alpha = .05$), 90% BCa percentile intervals (5%/95% bounds) are reported (Appendix B, Table 7). Performance expectancy was positively associated with behavioral intention (H2: $\beta = 0.585$, 90% BCa CI [0.480; 0.678]). Social influence (H3: $\beta = 0.179$, 90% BCa CI [0.115; 0.245]), price value (H6: $\beta = 0.179$, 90% BCa CI [0.096; 0.260]), and habit (H7: $\beta = 0.179$, 90% BCa CI [0.097; 0.265]) also showed positive associations with behavioral intention. In contrast, the BCa confidence intervals for effort expectancy (H1: $\beta = -0.051$, 90% BCa CI [−0.136; 0.023]), facilitating conditions (H4: $\beta = -0.027$, 90% BCa CI [−0.108; 0.046]), and hedonic motivation (H5: $\beta = -0.002$, 90% BCa CI [−0.109; 0.113]) included zero, providing no support for these hypotheses.

Effect sizes indicated a large effect of performance expectancy on behavioral intention ($f^2 = 0.630$). Social influence ($f^2 = 0.098$), price value ($f^2 = 0.092$), and habit ($f^2 = 0.072$) exhibited small effects, whereas all remaining predictors showed negligible effects (Appendix B, Table 10).

As a robustness check, a moderation model including age and gender (direct effects and interaction terms) was estimated (Appendix B, Table 19). Among the interaction terms, only age × price value showed a negative conditional effect on behavioral intention ($\beta = -0.164$, 90% BCa CI [−0.314; −0.071]), while the remaining interaction effects had BCa intervals spanning zero. Model comparison indicates that adding age and gender marginally increases explained variance

**Table 2.  Pearson correlation coefficients (\*\*\* p<.001, \*\* p<.01, \* p<.05).**

| No. | Construct | M | SD | (1) | (2) | (3) | (4) | (5) | (6) | (7) | (8) | (9) | (10) |
|---|---|---|---|---|---|---|---|---|---|---|---|---|---|
| (1) | Performance Expectancy | 5.49 | 1.40 | – | .535 \*\*\* | .237 \*\* | .668 \*\*\* | .466 \*\*\* | .375 \*\*\* | .255 \*\*\* | .412 \*\*\* | .082 | −.195 \*\* |
| (2) | Effort Expectancy | 5.68 | 1.32 | | – | .511 \*\*\* | .410 \*\*\* | .690 \*\*\* | .514 \*\*\* | .435 \*\*\* | .824 \*\*\* | −.135 \* | −.175 \*\* |
| (3) | Social Influence | 4.09 | 1.45 | | | – | .161 \* | .469 \*\*\* | .308 \*\*\* | .466 \*\*\* | .595 \*\*\* | −.070 | −.085 |
| (4) | Facilitating Conditions | 5.96 | 1.09 | | | | – | .361 \*\*\* | .313 \*\*\* | .129 \* | .298 \*\*\* | .055 | −.285 \*\*\* |
| (5) | Hedonic Motivation | 4.95 | 1.61 | | | | | – | .483 \*\*\* | .591 \*\*\* | .649 \*\*\* | −.052 | −.155 \* |
| (6) | Price Value | 5.02 | 1.88 | | | | | | – | .455 \*\*\* | .599 \*\*\* | −.165 \* | −.107 |
| (7) | Habit | 3.79 | 1.52 | | | | | | | – | .560 \*\*\* | −.058 | −.039 |
| (8) | Behavioral Intention | 5.22 | 1.64 | | | | | | | | – | −.162 \* | −.154 \* |
| (9) | Gender | – | – | | | | | | | | | – | .068 |
| (10) | Age | 24.69 | 4.62 | | | | | | | | | | – |

(ΔR² = +0.023) and slightly improves SRMR (ΔSRMR = −0.004), but reduces predictive performance (ΔQ²predict = −0.068; higher prediction errors for BI: ΔRMSE = +0.067, ΔMAE = +0.034; Appendix B, Table 20). The structural model results are summarized in Fig 4.

## 7 Discussion

This study examined determinants of behavioral intention to use an indoor navigation user interface mock-up in a university context using UTAUT2. The work contributes to IPS research by shifting attention from technical feasibility toward early-stage intention formation grounded in concrete interaction. Participants did not merely read a scenario, but completed task-based use cases within an interactive UI mock-up. This aligns the evaluation with an early design phase in which perceived value is shaped by interaction with a UI concept rather than by full technical performance.

Performance expectancy emerged as the strongest predictor of behavioral intention. This indicates that, in an indoor navigation context, intention is primarily driven by perceived functional value and the expectation that the system helps accomplish goals efficiently. The finding is consistent with prior work in related location-based service contexts and broader UTAUT2 syntheses [17,47,54]. It also aligns with acceptance research on outdoor navigation and wayfinding systems [28–30]. For IPS UI design, the result suggests that "navigation" alone may not be sufficient to create strong perceived performance. Instead, performance gains should be made tangible through features that reduce uncertainty and time cost in complex buildings. Accordingly, value may extend beyond route guidance through context-relevant functions (e.g., searching for people/rooms, locating services, planning), and through information that reduces waiting and detours (e.g., emergency rooms or airport checkpoints [34,35]) or cafeteria crowding and service availability [10]. The practical implication is not only to implement such functions, but also to communicate them clearly so that users can anticipate concrete benefits before first use.

Price value was positively related to behavioral intention. This differs from some LBS findings where price value may be weaker when users are accustomed to free navigation apps, as argued in tourism settings [19]. In the present study, price value was elicited under the specific pricing frame of a one-time fee of €0.99. Under this frame, participants perceived benefits as outweighing costs. At the same time, price value should be interpreted as conditional on the communicated pricing scheme. Because price value is a benefit–cost trade-off, different deployment models (institutional provision,

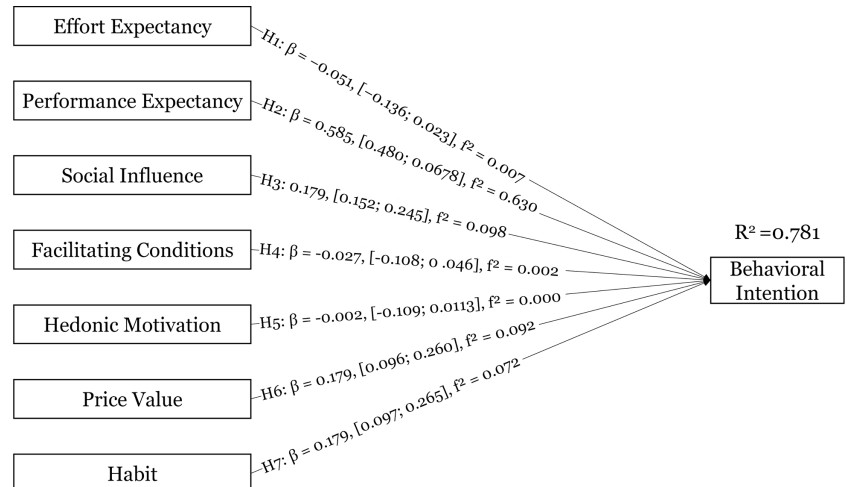

**Fig 4. Research model results.** Standardized path coefficients (β) with 90% BCa bootstrap confidence intervals and effect sizes (f²).

bundled services, subscriptions, free access) may change its salience [13]. Pricing-related implications should therefore be framed as a boundary condition and tested under alternative pricing models [19].

Habit and social influence also showed positive relationships with behavioral intention. Habit is commonly interpreted as an outcome of repeated use over time [13]. In a short-term mock-up study, however, a significant habit path likely captures a predisposition toward routinized use of navigation tools rather than true habit formation with the evaluated system. Longitudinal field deployments are needed to test whether repeated IPS use produces stable habit effects.

Because the main analysis pooled three independent data collections, mean differences across collections were examined using a one-way MANOVA on the study constructs (see Appendix G). The multivariate test did not indicate systematic differences across samples at $\alpha = .05$ (Wilks' $\Lambda = 0.868$, F(16, 342) = 1.572, p = .074). Follow-up univariate tests showed no differences for BI and the remaining UTAUT2 constructs (all p > .05), whereas habit differed across data collections (F(2, 178) = 4.399, p = .014, $\eta p^2 = 0.047$). Tukey post-hoc tests indicate higher habit in the smaller Clickworker sample (M = 4.67) compared to the university sample (M = 3.38) and the larger Clickworker sample (M = 3.77). This baseline heterogeneity provides a boundary condition for interpreting the habit path in the pooled structural model.

Social influence supports the view that recommendations, endorsements, and perceived expectations of relevant others can shape intention in emerging navigation contexts when direct experience is limited [13]. Evidence from IPS-related healthcare contexts also points to the relevance of social norms [15]. In contrast, some tourism LBS research reports weaker social influence effects [19], which may reflect higher familiarity with outdoor navigation apps and lower reliance on social cues. Because IPS deployments remain comparatively rare in practice [3], social cues and institutional endorsement may serve as signals of legitimacy. Early rollouts may therefore benefit from awareness and credibility building (communication, institutional integration, visible endorsement) in addition to feature delivery.

Effort expectancy, facilitating conditions, and hedonic motivation were not supported because their one-tailed 90% BCa confidence intervals include zero. In a brief, guided mock-up interaction, effort expectancy may show ceiling effects (limited variance), while facilitating conditions may become salient only in real deployments where infrastructure and organizational support constraints are experienced. Hedonic motivation may be less relevant in utilitarian university navigation but could matter more in leisure-oriented indoor settings.

The model explains a substantial portion of variance in behavioral intention. High explained variance can occur in homogeneous, single-behavior settings, but it can also raise concerns about overlap and method bias. The analysis therefore included collinearity checks and a common method assessment using full collinearity VIFs. Inner-model VIFs were low and full-collinearity VIFs remained below 5.0, suggesting no severe distortion. The findings should nonetheless be interpreted as context-specific to an early-stage UI mock-up evaluation, not as a general claim that IPS adoption can routinely be explained with similarly high accuracy across settings.

A robustness analysis estimated the UTAUT2 moderation structure with age and gender. The pattern of main effects remained stable. Only the age × price value interaction showed a negative conditional effect, indicating that the role of price value weakens with increasing age in this sample. Given the sample's age distribution, this moderation should be treated as exploratory; adding moderators did not improve predictive performance.

Overall, the results suggest that early-stage intention toward an IPS UI concept is driven most strongly by performance-related beliefs, complemented by perceived value, social endorsement, and predispositions toward routinized navigation-tool use. The findings should be interpreted as guidance for prototype-based UI evaluation and prioritization, not as evidence about the adoption of fully functional IPS deployments that include localization accuracy, latency, navigation errors, and other technical frictions.

## 8 Conclusion

This study applied UTAUT2 to examine determinants of behavioral intention to use an indoor navigation user interface (UI) mock-up in a university context [13] using a pooled sample of 181 potential users from three independent data collections.

The first research question was addressed by designing literature-based use cases and translating them into task flows integrated into an interactive UI mock-up. The second research question was addressed by identifying which UTAUT2 determinants are most closely associated with intention to use this prototype-based IPS concept after task-based interaction with the mock-up. Overall, the findings provide evidence on early-stage intention formation toward an IPS UI concept and offer guidance for prototype-oriented design decisions in comparable indoor navigation contexts, but should not be interpreted as evidence on the adoption of fully functional IPS deployments under real technical constraints.

### 8.1 Theoretical implications

This study contributes to technology acceptance and IPS-related literature in four ways.

First, it extends UTAUT2 application to an IPS-related context in which evaluation is anchored in task-based interaction with an interactive UI mock-up. This complements prior IPS UI work that often focuses on interface design and technical feasibility in specific contexts, such as hospitals and universities [10,11,16,27] by adding a theory-based explanation of intention formation, which has been argued as relevant for successful deployment decisions [2].

Second, the results clarify which UTAUT2 determinants are most salient for intention toward an indoor navigation UI concept in the evaluated context [13]. Performance expectancy emerged as the strongest driver, complemented by social influence, price value (under the communicated one-time fee of €0.99), and habit. At the same time, effort expectancy, facilitating conditions, and hedonic motivation showed no support in this setting. These findings inform acceptance mechanisms in prototype-focused, short-term evaluations and help align early-stage IPS UI evaluation with established acceptance theory.

Third, the study provides a structured set of indoor navigation use cases and task flows implemented in a UI mock-up. This constitutes a reusable artifact for future research that aims to evaluate indoor navigation concepts in a controlled manner. While the artifact was developed for a university setting, the general logic of task-based navigation and information needs can be adapted to other indoor environments, provided that context-specific constraints and user groups are considered.

Fourth, the robustness analysis that included the standard UTAUT2 moderators age and gender supports the stability of the main-effects pattern. Only the age × price value interaction indicated a negative moderation, whereas the remaining interaction effects were not supported, and the added complexity did not improve predictive performance. This supports interpreting the core model as the primary specification for this dataset, while treating moderation as supplementary evidence rather than a central contribution.

### 8.2 Practical implications

The results offer practical implications for early-stage IPS UI design and evaluation.

First, prototype concepts for indoor navigation should prioritize performance-related value propositions and make them tangible through features that reduce time cost, uncertainty, and search effort. Beyond routing, context-relevant informational features can strengthen perceived usefulness in specific indoor settings, consistent with prior IPS UI research emphasizing the importance of context-specific functionality and task support.

Second, social influence suggests that institutional endorsement, communication, and visibility may matter in early rollout phases [13]. Accordingly, stakeholders such as universities, hospitals, or companies should consider credibility-building (visibility, endorsement, communication) alongside feature delivery before large-scale deployment.

Third, pricing implications should be interpreted strictly within the communicated pricing frame used in the study. Price value was measured under a one-time fee of €0.99. Alternative deployment models, such as institutional provision, subscriptions, or free access, may alter perceived value and the relative importance of adoption drivers [17]. Pricing and rollout decisions should therefore be validated through context-specific field trials or experiments that vary price levels and pricing models.

Overall, these implications concern prototype-based UI concept decisions and prioritization. They do not constitute evidence for adoption outcomes of fully functional IPS deployments under real technical constraints.

### 8.3 Limitations and future research

Several limitations restrict the scope of the conclusions and motivate future research.

First, the study measures self-reported behavioral intention rather than observed usage behavior. The task-based UI mock-up captures short-term evaluations in a compensated crowdworking setting and cannot represent sustained adoption, real-world constraints, or behavioral consequences. Future research should validate the model in field deployments using behavioral data, such as app logs, and longitudinal designs that capture continued use and post-adoption dynamics.

Second, the mock-up did not expose participants to IPS-specific technical frictions such as localization inaccuracies, signal instability, latency, navigation errors, connectivity constraints, or battery drain. Because these factors can substantially shape user experience and trust, intention ratings may be upward-biased compared to real deployments. Future studies should incorporate realistic performance conditions and error cases, ideally through functional prototypes tested in situ.

Third, the sample was collected in Germany via a crowdworking platform and is skewed toward younger participants. Together with strict data-quality criteria and the resulting exclusion rate, this may introduce selection effects and limit generalizability. Replications should target more diverse samples and compare indoor environments that differ in complexity, stress, and time pressure. To assess whether pooling across the three independent data collections is plausible, a one-way MANOVA (see Appendix G) indicated no systematic mean differences across the UTAUT2 constructs and BI, except for habit, which was higher in the smaller Clickworker sample. This suggests that pooling is broadly defensible, but the habit effect should be interpreted cautiously and replicated in larger, more balanced samples. In addition, future work can test whether results replicate consistently across the separate data collections (e.g., via invariance or multi-group comparisons).

Fourth, the model explains a substantial share of variance in behavioral intention. High explained variance can occur in homogeneous, belief-based models focused on a single target behavior. At the same time, it can be inflated by conceptual overlap, collinearity, or common method influences. Collinearity and common method diagnostics were therefore examined following established guidance, and conclusions should remain bound to the prototype-based setting and measurement approach. Future research should further reduce single-method dependence through multi-source designs, behavioral measures, and experimental manipulations.

Fifth, price value was assessed under a specific communicated price frame. Future studies should experimentally vary pricing levels and deployment models to test boundary conditions and to determine when price value becomes decisive for adoption decisions.

### Information on informed consent of participants

At the beginning of our questionnaire, we gathered a written consent of our participants that they agree that their data is used for scientific purposes, as this was the prerequisite for them to participate.

### Author contributions

**Conceptualization:** Thomas Paetow, Michael Leyer.

**Data curation:** Thomas Paetow, Johannes Wichmann, Michael Leyer.

**Formal analysis:** Johannes Wichmann.

**Investigation:** Thomas Paetow, Johannes Wichmann, Michael Leyer.

**Methodology:** Thomas Paetow, Johannes Wichmann, Michael Leyer.

**Project administration:** Thomas Paetow.

**Software:** Thomas Paetow.

**Supervision:** Michael Leyer.

**Validation:** Johannes Wichmann, Michael Leyer.

**Visualization:** Thomas Paetow.

**Writing – original draft:** Thomas Paetow, Johannes Wichmann.

**Writing – review & editing:** Michael Leyer.

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
