## [Decision Letter · Decision Letter 0]

16 Dec 2025

PONE-D-25-01491Why Would Individuals Use Indoor Positioning Systems? A Study from a Potential User PerspectivePLOS One?

Dear Dr. Wichmann,

Thank you for submitting your manuscript to PLOS ONE. After careful consideration, we feel that it has merit but does not fully meet PLOS ONE’s publication criteria as it currently stands. Therefore, we invite you to submit a revised version of the manuscript that addresses the points raised during the review process.

**1. Conceptual Validity: Prototype vs. Actual System (Crucial)**

Both reviewers, particularly Reviewer 2, identify a fundamental disconnect between the study's claims and its stimulus.

**The Issue:** You evaluated a static/interactive User Interface (UI) mock-up, not a functional Indoor Positioning System (IPS). Real IPS usage involves signal latency, localization errors, and battery drain, none of which were experienced by users.**The Requirement:** You must significantly qualify your findings. You cannot claim to measure "IPS adoption" in general. You are measuring the acceptance of a UI concept for indoor navigation. The manuscript must explicitly discuss this limitation and how the lack of technical friction (e.g., navigation errors) might inflate acceptance scores.

**2. Data Quality and Sampling Rigor**

There are serious concerns regarding how the data was processed and the final sample size.

**High Exclusion Rate:** You excluded 268 out of 392 completed questionnaires (approx. 68% rejection rate). This is exceptionally high and suggests potential flaws in the survey design, task complexity, or attention checks. You must provide a detailed analysis of why so many participants failed.**Sample Size:** The final sample (<code>N</code><code _ngcontent-ng-c1097911779=" " class="rendered">=123)</code> is considered small for a UTAUT2 model with moderators (Reviewer 1). You need to justify the statistical power of this sample size more robustly.**Demographics:** The sample is skewed toward young German males/students. You must discuss how this limits generalizability to other populations (e.g., hospital visitors, elderly users).

**3. Statistical Anomalies and Model Fit**

The statistical results raise red flags regarding validity.

**Suspiciously High R2****:** An explained variance of 0.859 is unusually high for behavioral intention studies (Reviewer 2). This suggests potential Common Method Bias (CMB), multicollinearity, or conceptual overlap between predictors. You must conduct and report rigorous tests for CMB and multicollinearity.**Insignificant Core Constructs:** Many core UTAUT2 variables (Effort Expectancy, Facilitating Conditions, Hedonic Motivation, Habit) were non-significant. The discussion currently lacks a deep theoretical explanation for why these established predictors failed in this context.**Missing Data:** Reviewer 1 notes that the specific list of items (especially for Spatial Abilities) and standard validity/reliability tables are missing or insufficient. These must be added.

**4. Theoretical Weaknesses: Spatial Abilities & Price Value**

The theoretical additions to the model are viewed as weak or poorly operationalized.

**Spatial Abilities (Hypothesis 8):** Both reviewers criticize the inclusion of Spatial Abilities. It was found to be insignificant, yet the discussion does not adequately explain why. The operational definition of this construct is unclear, and its theoretical integration needs to be much stronger—or reconsidered.**Price Value:** Setting a fixed hypothetical price (€0.99) is problematic (Reviewer 2). In a university context, students often expect free services. This framing likely skewed the results and needs to be discussed as a limitation.

**5. Structure and Presentation**

**Introduction:** Remove methodological details (sample size, specific stats) from the Introduction; focus on the theoretical gap.**Comparison:** The literature review should better contrast outdoor vs. indoor navigation prototypes to justify the specific design choices made.

We look forward to receiving your revised manuscript.

Kind regards,

Frantisek Sudzina

Academic Editor

PLOS One

Journal Requirements:

Additional Editor Comments:

1. Conceptual Validity: Prototype vs. Actual System (Crucial)

Both reviewers, particularly Reviewer 2, identify a fundamental disconnect between the study's claims and its stimulus.

The Issue: You evaluated a static/interactive User Interface (UI) mock-up, not a functional Indoor Positioning System (IPS). Real IPS usage involves signal latency, localization errors, and battery drain, none of which were experienced by users.

The Requirement: You must significantly qualify your findings. You cannot claim to measure "IPS adoption" in general. You are measuring the acceptance of a UI concept for indoor navigation. The manuscript must explicitly discuss this limitation and how the lack of technical friction (e.g., navigation errors) might inflate acceptance scores.

2. Data Quality and Sampling Rigor

There are serious concerns regarding how the data was processed and the final sample size.

High Exclusion Rate: You excluded 268 out of 392 completed questionnaires (approx. 68% rejection rate). This is exceptionally high and suggests potential flaws in the survey design, task complexity, or attention checks. You must provide a detailed analysis of why so many participants failed.

Sample Size: The final sample (N=123) is considered small for a UTAUT2 model with moderators (Reviewer 1). You need to justify the statistical power of this sample size more robustly.

Demographics: The sample is skewed toward young German males/students. You must discuss how this limits generalizability to other populations (e.g., hospital visitors, elderly users).

3. Statistical Anomalies and Model Fit

The statistical results raise red flags regarding validity.

Suspiciously High R2: An explained variance of 0.859 is unusually high for behavioral intention studies (Reviewer 2). This suggests potential Common Method Bias (CMB), multicollinearity, or conceptual overlap between predictors. You must conduct and report rigorous tests for CMB and multicollinearity.

Insignificant Core Constructs: Many core UTAUT2 variables (Effort Expectancy, Facilitating Conditions, Hedonic Motivation, Habit) were non-significant. The discussion currently lacks a deep theoretical explanation for why these established predictors failed in this context.

Missing Data: Reviewer 1 notes that the specific list of items (especially for Spatial Abilities) and standard validity/reliability tables are missing or insufficient. These must be added.

4. Theoretical Weaknesses: Spatial Abilities & Price Value

The theoretical additions to the model are viewed as weak or poorly operationalized.

Spatial Abilities (Hypothesis 8): Both reviewers criticize the inclusion of Spatial Abilities. It was found to be insignificant, yet the discussion does not adequately explain why. The operational definition of this construct is unclear, and its theoretical integration needs to be much stronger—or reconsidered.

Price Value: Setting a fixed hypothetical price (€0.99) is problematic (Reviewer 2). In a university context, students often expect free services. This framing likely skewed the results and needs to be discussed as a limitation.

5. Structure and Presentation

Introduction: Remove methodological details (sample size, specific stats) from the Introduction; focus on the theoretical gap.

Comparison: The literature review should better contrast outdoor vs. indoor navigation prototypes to justify the specific design choices made.

Reviewers' comments:

Reviewer's Responses to Questions

**Comments to the Author**

1. Is the manuscript technically sound, and do the data support the conclusions?

Reviewer #1: Partly

Reviewer #2: No

2. Has the statistical analysis been performed appropriately and rigorously?

Reviewer #1: No

Reviewer #2: No

3. Have the authors made all data underlying the findings in their manuscript fully available?

Reviewer #1: No

Reviewer #2: No

4. Is the manuscript presented in an intelligible fashion and written in standard English?

Reviewer #1: No

Reviewer #2: Yes

Reviewer #1: It is recommended to adhere to the academic writing style for a quantitative study, namely by refraining from utilising terms such as 'we' or 'our study' or ‘they’ in the text.

Abstract

The overall sample of 123 is insufficient for survey methods and does not align with the claim that IPS has gained popularity. The moderator should be highlighted in the abstract as a major contribution of this study.

Introduction

The research problem appears insufficiently robust to warrant the application of UTAUT2. Researchers employ UTAUT to understand the factors that influence both intentions and actual behaviour when using technology. However, the problem articulated primarily concentrated on technological or system design concerns.

The content should be realigned with the sub-title; for instance, in the Introduction section, the sample size and statistical methods should not be reiterated, as they are already covered in the abstract and subsequently in the methodology section. The study's significance focusses primarily on application, neglecting theoretical contributions.

Literature Review – Indoor Positioning Systems

The discussion primarily emphasises that previous studies fail to address the aspect of intention to use. Given that the technology purports to possess a distinctive system, a robust rationale must be articulated regarding the necessity of employing the theory in relation to this unique system.

Literature review – Developing an IPS prototype

Since the author(s) have noted the differences in navigation outdoors. This section must compare the two prototypes to provide a clear description.

Literature review – theory for evaluation

Similar to the previous discussion, it indicated that the theory has not been tested in the IPS prototype. No novel components have been incorporated into the theory, owing to the distinctive prototype of the technology discussed in prior sections. The requirements of the moderator within the theory should be emphasised in the discourse, along with the rationale for including spatial abilities within the theory. The points may be incorporated in the introduction as research issues.

Literature Review – Hypotheses & Research Model

The discussion appears to have led to a superficial hypothesis. Given the significance of Hypothesis 8 to the theory, the debate necessitates the strengthening and enhancement of Hypothesis 8. There is a lack of comprehensive reviews on topics related to spatial ability.

Method

Out of 464 questionnaires, only 26.51% were deemed useful, indicating either a flawed sample selection process or a less accurate questionnaire. The list of items to measure the variables should be presented, especially the spatial abilities. Missing of few statistical test: descriptive, reliability, validity, VIF etc.

Results and conclusion

The operational definition of spatial abilities has been overlooked, resulting in the diminished significance of this study's primary contribution.

The primary focus of this study should be on analysing the moderator, or spatial ability. However, the insignificance of the moderators has raised numerous doubts. The author(s) should concentrate on justifying the potential insignificance of the moderator within the framework. Regrettably, the limitation did not address the practical definition of the moderators or their potential insignificance.

Reviewer #2: This manuscript addresses a relevant and timely topic by investigating individuals’ intention to use indoor positioning systems (IPS). The study is carefully structured and methodologically transparent; however, several substantial conceptual and methodological limitations need to be addressed before the conclusions can be considered robust.

First, a central limitation of the study is that it does not evaluate an actual indoor positioning system but rather an interactive user interface prototype without real positioning technology. Critical characteristics of IPS—such as localization accuracy, signal instability, latency, or navigation errors—are not part of the user experience. Nevertheless, the manuscript repeatedly draws conclusions about the adoption and use of “IPS” in general. This conceptual leap risks overstating the implications of the findings and should be substantially qualified.

Second, although the authors attempt to reduce the intention–behavior gap by requiring interaction with the prototype, the study still relies exclusively on self-reported behavioral intention in an artificial and short-term setting. Participants complete predefined tasks in a mock-up environment under monetary incentives, which does not reflect real-world usage, sustained adoption, or behavioral consequences. As a result, the external and ecological validity of the findings remains limited.

Third, the sampling strategy raises concerns. Of the 392 completed questionnaires, 268 were excluded due to failed control questions, resulting in a final sample of only 123 participants. This exceptionally high exclusion rate warrants a more critical discussion, as it may indicate issues related to task complexity, participant engagement, or survey design. Furthermore, the sample is heavily skewed toward young, male, German university members, which severely limits generalizability to other user groups, cultural contexts, and indoor environments such as hospitals or airports.

Fourth, the reported explained variance of behavioral intention (R² = 0.859) is unusually high for technology acceptance research. Such a value may point to conceptual overlap between predictors, common method bias, or an overly context-specific model. The manuscript does not sufficiently reflect on this issue, nor does it discuss potential inflation of explanatory power.

Fifth, several core UTAUT2 constructs—effort expectancy, facilitating conditions, hedonic motivation, and habit—do not show significant effects. While null findings are reported, their theoretical implications are not deeply explored. Similarly, the inclusion of spatial abilities as a moderating variable appears weakly justified and empirically unsupported, raising questions about its conceptual integration within the model.

Finally, the operationalization of price value through a fixed hypothetical price (€0.99) introduces a strong framing effect. This price does not reflect realistic IPS deployment models (e.g., institutional provision, subscription-based services, or free access), limiting the interpretability and transferability of the observed price value effect.

In summary, while the study demonstrates methodological diligence and addresses an underexplored area, its conclusions should be more cautiously framed, and several conceptual, methodological, and interpretative issues require substantial revision.

**Do you want your identity to be public for this peer review?** For information about this choice, including consent withdrawal, please see our Privacy Policy

Reviewer #1: No

Reviewer #2: No

---

## [Author Response · Author response to Decision Letter 1]

11 Feb 2026

PONE-D-25-01491

Why Would Individuals Use Indoor Positioning Systems? A Study from a Potential User Interface Perspective

Point-by-Point-Response to the Reviewers

Introduction

Dear Editor, dear Reviewers,

Thank you for the time and effort you invested in reviewing our manuscript. We appreciate your thoughtful comments and constructive suggestions, which helped us improve the clarity, rigor, and positioning of the paper. We have revised the manuscript accordingly and addressed each point below.

In the following, each comment is reproduced and answered in turn, with the corresponding changes indicated in the manuscript.

1 Response to the Editor

ID: 1.1

Suggestion for improvement

Conceptual Validity: Prototype vs. Actual System (Crucial)

Both reviewers, particularly Reviewer 2, identify a fundamental disconnect between the study's claims and its stimulus.

The Issue: You evaluated a static/interactive User Interface (UI) mock-up, not a functional Indoor Positioning System (IPS). Real IPS usage involves signal latency, localization errors, and battery drain, none of which were experienced by users.

The Requirement: You must significantly qualify your findings. You cannot claim to measure "IPS adoption" in general. You are measuring the acceptance of a UI concept for indoor navigation. The manuscript must explicitly discuss this limitation and how the lack of technical friction (e.g., navigation errors) might inflate acceptance scores.

Answer

Thank you for highlighting this conceptual scope issue. We revised the manuscript to consistently frame the study as an early-stage evaluation of behavioral intention toward an interactive IPS UI mock‑up, rather than adoption of a fully functional IPS. The abstract, introduction, discussion, and limitations now explicitly state that the stimulus is a prototype UI concept and therefore does not include deployment-specific technical frictions (e.g., localization errors, latency), which may upward-bias intention. Accordingly, implications are formulated as prototype-based guidance (UI feature prioritization and value communication), not as evidence on real IPS adoption. See also responses to Reviewer 2 (IDs 3.1 and 3.2).

Change

see Abstract: added explicit scope qualification and prototype framing.

see Introduction: clarified that the study evaluates intention after task-based interaction with a UI mock‑up and bounded claims accordingly.

see Discussion / Conclusion (Limitations): emphasized intention-behavior gap and missing real deployment frictions.

ID: 1.2

Suggestion for improvement

Data Quality and Sampling Rigor

There are serious concerns regarding how the data was processed and the final sample size.

High Exclusion Rate: You excluded 268 out of 392 completed questionnaires (approx. 68% rejection rate). This is exceptionally high and suggests potential flaws in the survey design, task complexity, or attention checks. You must provide a detailed analysis of why so many participants failed.

Sample Size: The final sample (N=123) is considered small for a UTAUT2 model with moderators (Reviewer 1). You need to justify the statistical power of this sample size more robustly.

Demographics: The sample is skewed toward young German males/students. You must discuss how this limits generalizability to other populations (e.g., hospital visitors, elderly users).

Answer

We agree that the high exclusion rate and sampling implications require transparent reporting and justification. The numbers were indeed unclear. Thus, we expanded the Participants and Data Collection section to clearly describe recruitment via Clickworker, the predefined data-quality protocol (including code-based task completion and consistency checks), and the participant flow. In addition, we collected further data using the same protocol and pooled the independent data collections to increase the number of valid cases (final pooled sample: N = 181). Finally, we added limitations explicitly addressing potential selection effects due to strict screening and the resulting constraints on generalizability. See also Reviewer 2 ID 3.3 and Editor ID 1.3 regarding robustness/diagnostics.

Change

see Method 5.1 / 5.2: clarified task-based procedure and code-based attention checks.

see Method 5.3: expanded participant flow, exclusion logic, and sample description.

see Limitations: added paragraph on representativeness and selection effects from strict data-quality criteria.

see Appendix G: MANOVA Results.

ID: 1.3

Suggestion for improvement

Statistical Anomalies and Model Fit

The statistical results raise red flags regarding validity.

Suspiciously High R2: An explained variance of 0.859 is unusually high for behavioral intention studies (Reviewer 2). This suggests potential Common Method Bias (CMB), multicollinearity, or conceptual overlap between predictors. You must conduct and report rigorous tests for CMB and multicollinearity.

Insignificant Core Constructs: Many core UTAUT2 variables (Effort Expectancy, Facilitating Conditions, Hedonic Motivation, Habit) were non-significant. The discussion currently lacks a deep theoretical explanation for why these established predictors failed in this context.

Missing Data: Reviewer 1 notes that the specific list of items (especially for Spatial Abilities) and standard validity/reliability tables are missing or insufficient. These must be added.

Answer

Thank you for raising concerns about unusually high explained variance and potential statistical artefacts. We revised the reporting and added robustness diagnostics. The model now explains a substantial but more plausible share of intention variance (R² = 0.781) and focusses on the core constructs. We report (i) collinearity diagnostics for the structural model predictors (VIF < 3), and (ii) a common method assessment using Kock’s full-collinearity approach (full collinearity VIFs ranged 2.48–4.56; several exceed 3.3, but all are below 5). We also report SRMR and predictive assessments (PLSpredict, CVPAT) and interpret model fit indices cautiously. In the discussion and limitations, we explicitly bound interpretation to the prototype-based setting and discuss why some UTAUT2 constructs were not supported. See also Reviewer 2 ID 3.4 and ID 3.5.

Change

see Results: updated R² and structural paths; clarified inference approach (BCa bootstrap, one-tailed).

see Method 5.4 / Appendix B: added full-collinearity VIFs and additional diagnostic reporting (SRMR, PLSpredict, CVPAT).

see Discussion / Limitations: added interpretation of high R² and null findings as context-specific.

ID: 1.4

Suggestion for improvement

Theoretical Weaknesses: Spatial Abilities & Price Value

The theoretical additions to the model are viewed as weak or poorly operationalized.

Spatial Abilities (Hypothesis 8): Both reviewers criticize the inclusion of Spatial Abilities. It was found to be insignificant, yet the discussion does not adequately explain why. The operational definition of this construct is unclear, and its theoretical integration needs to be much stronger—or reconsidered.

Price Value: Setting a fixed hypothetical price (€0.99) is problematic (Reviewer 2). In a university context, students often expect free services. This framing likely skewed the results and needs to be discussed as a limitation.

Answer

We addressed both theoretical concerns. First, we removed Spatial Abilities and the associated hypotheses to keep the model aligned with canonical UTAUT2 and to avoid introducing a non-standard moderator without strong theoretical grounding. Second, we clarified that Price Value is conditional on the communicated pricing frame used in the study (one-time fee of €0.99) and therefore must be interpreted as a boundary condition rather than as a general pricing conclusion for IPS. The discussion and conclusion now explicitly state that alternative deployment models (institutional provision, subscriptions, free access) may alter price-related evaluations and should be tested in future research. See also Reviewer 1 ID 2.7 and Reviewer 2 ID 3.6.

Change

see Theory / Hypotheses / Research model: removed Spatial Abilities construct and all related hypotheses.

see Method 5.1: clarified Price Value measurement under the €0.99 one-time fee frame.

see Discussion / Conclusion: framed pricing implications cautiously as pricing-frame dependent.

ID: 1.5

Suggestion for improvement

Structure and Presentation

Introduction: Remove methodological details (sample size, specific stats) from the Introduction; focus on the theoretical gap.

Comparison: The literature review should better contrast outdoor vs. indoor navigation prototypes to justify the specific design choices made.

Answer

We revised the manuscript structure to improve clarity and theoretical positioning. The introduction now focuses on the research gap (behavioral intention toward IPS UI concepts) and states the study’s scope and contributions, while methodological detail is confined to the Method section. In addition, we added a dedicated subsection contrasting indoor vs. outdoor navigation acceptance research to clarify why OPS findings do not transfer directly to IPS UI contexts. Overall, the narrative now follows a clearer logic from (i) IPS context and use cases → (ii) UTAUT2-based model → (iii) task-based mock‑up evaluation → (iv) results and bounded implications. See also Reviewer 1 IDs 2.3 and 2.5.

Change

see Introduction: strengthened gap/contribution framing; reduced procedural detail.

see (new) Section 2.2: added OPS behavioral research contrast and transfer-limit argument.

2 Response to Review(er) 1

ID: 2.1

Suggestion for improvement

It is recommended to adhere to the academic writing style for a quantitative study, namely by refraining from utilising terms such as 'we' or 'our study' or ‘they’ in the text

Answer

We revised the manuscript to use a more formal academic style by reducing first-person phrasing and using neutral formulations (e.g., “this study”, passive voice) where appropriate. We also adjusted literature descriptions to avoid informal “they” phrasing and instead refer directly to authors/studies.

Change

All chapters: editing pass to reduce personal pronouns and informal wording.

ID: 2.2

Suggestion for improvement

Abstract | The overall sample of 123 is insufficient for survey methods and does not align with the claim that IPS has gained popularity. The moderator should be highlighted in the abstract as a major contribution of this study.

Answer

Thank you for this helpful comment. We updated the abstract accordingly. The sample size is now reported as a pooled sample of N = 181 collected across independent data collections using the same protocol. We also revised the wording on IPS prevalence to avoid overstatement and instead note that real-world deployments remain comparatively scarce. Moderation (age, gender) is reported as a robustness analysis (UTAUT2). Most interaction effects were not supported because their BCa confidence intervals include zero. Only the Age × Price Value interaction showed a negative effect (see Appendix B, Table 19). Because adding moderators did not improve predictive performance, moderation it was removed from the main analysis as suggested by the AE. Thus, is documented as robustness evidence rather than positioned as the main contribution in the abstract.

Change

see Abstract: updated sample size (N=181), adjusted IPS deployment wording, and clarified scope.

see Method / Appendix: moderation robustness analysis documented.

ID: 2.3

Suggestion for improvement

Introduction | The research problem appears insufficiently robust to warrant the application of UTAUT2. Researchers employ UTAUT to understand the factors that influence both intentions and actual behaviour when using technology. However, the problem articulated primarily concentrated on technological or system design concerns.

The content should be realigned with the sub-title; for instance, in the Introduction section, the sample size and statistical methods should not be reiterated, as they are already covered in the abstract and subsequently in the methodology section. The study's significance focusses primarily on application, neglecting theoretical contributions.

Answer

We revised the introduction to strengthen the research problem and theoretical contribution. The introduction now emphasizes the gap that IPS research often focuses on technical feasibility, while evidence on adoption drivers and intention formation grounded in UI interaction remains limited. We also state the study’s contributions (use-case-driven UI mock‑up + task-based evaluation + UTAUT2 determinants) and keep methodological specifics (sample size, analysis technique) in the Method section. The research questions are now explicitly stated.

Change

see Introduction: rewritten gap statement, contributions, and explicit research questions; removed methodological detail from the introduction.

ID: 2.4

Suggestion for improvement

Literature Review – Indoor Positioning Systems | The discussion primarily emphasises that previous studies fail to address the aspect of intention to use. Given that the technology purports to possess a distinctive system, a robust rationale must be articulated regarding the necessity of employing the theory in relation to this unique system.

Answer

We expanded the theoretical background to better cover IPS-related research and to justify the choice of UTAUT2 for evaluating intention toward an IPS UI concept. The revised background now integrates IPS use cases and UI prototyping literature and positions UTAUT2 as suitable for voluntary, consumer-like, early-stage adoption evaluations.

Change

see Sections 2–4: expanded IPS/UI background and LBS adoption evidence; clarified rationale for UTAUT2.

ID: 2.5

Suggestion for improvement

Literature review – Developing an IPS prototype | Since the author(s) have noted the differences in navigation outdoors. This section must compare the two prototypes to provide a clear description.

Answer

We addressed this by adding a dedicated subsection on outdoor navigation (OPS) behavioral research and explicitly contrasting OPS vs. IPS in constraints, technologies, and typical use cases. This section clarifies why OPS acceptance findings cannot be transferred directly to IPS UI concepts and motivates the need for IPS-specific intention research.

Change

see Section 2.2: added OPS acceptance studies and an explicit indoor-outdoor contrast argument.

ID: 2.6

Suggestion for improvement

Literature review – theory for evaluation | Similar to the previous discussion, it indicated that the theory has not been tested in the IPS prototype. No novel components have been incorporated into the theory, owing to the distinctive prototype of the technology discussed in prior sections. The requirements of the moderator within the theory should be emphasised in the discourse, along with the rationale for including spatial abilities within the theory. The points may be incorporated in the introduction as research issues.

Answer

We revised the model to align with UTAUT2 and added the standard moderators age and gender as a robustness analysis (direct effects and interactions with all predictors). We report the moderation model output and compare model performance with the main model in the Appendix. In addition, the previously included Spatial Abilities construct was removed (see ID 2.7).

Change

see Method 5.4 / Appendix B: added moderation robustness analysis (age, gender) and performance comparison.

see Research model: Spatial Abilities removed.

ID: 2.7

Suggestion for improvement

Literature Review – Hypotheses & Research Model | The discussion appears to have led to a superficial hypothesis. Given the significance of Hypothesis 8 to the theory, the debate necessitates the strengthening and enhancement of Hypothesis 8. There is a lack of comprehensive reviews on topics related to spatial ability.

Answer

We agree and removed the Spatial Abilities construct and all associated hypotheses (formerly H8.*). This keeps the model consistent with UTAUT2 and avoids introducing a moderator without strong theoretical grounding in this context.

Change

see Hypo

---

## [Editor Report · Decision Letter 1]

16 Feb 2026

Why Would Individuals Use Indoor Positioning Systems? A Study from a Potential User Interface Perspective

PONE-D-25-01491R1

Dear Dr. Wichmann,

We’re pleased to inform you that your manuscript has been judged scientifically suitable for publication and will be formally accepted for publication once it meets all outstanding technical requirements.

Kind regards,

Frantisek Sudzina

Academic Editor

PLOS One
---

## [Editor Report · Acceptance letter]

PONE-D-25-01491R1

PLOS One

Dear Dr. Wichmann,

I'm pleased to inform you that your manuscript has been deemed suitable for publication in PLOS One. Congratulations! Your manuscript is now being handed over to our production team.

Kind regards,

on behalf of

Dr. Frantisek Sudzina

Academic Editor

PLOS One